# Effect of Health Education Intervention on Knowledge and Adherence to Intermittent Preventive Treatment of Malaria in Pregnancy Among Women

**DOI:** 10.3390/healthcare13020105

**Published:** 2025-01-08

**Authors:** Pauline N. Atser, Gommaa Hayat, Uchenna B. Okafor

**Affiliations:** 1Department of Nursing Sciences, Faculty of Basic Medical Sciences, College of Health Sciences, Benue State University, Makurdi 970101, Nigeria; atserpauline2013@gmail.com; 2Department of Nursing Sciences, Faculty of Allied Health Sciences, Ahmadu Bello University, Zaria 810211, Nigeria; h_gommaa@abu.edu.ng; 3Department of Nursing Sciences, Faculty of Basic Medical and Health Sciences, Walter Sisulu University, Nelson Mandela Drive Campus, Mthatha 5117, South Africa

**Keywords:** malaria treatment and prevention, health education intervention, knowledge, pregnancy

## Abstract

**Aim:** Malaria in pregnancy is a global health problem because it causes anemia in the mother and may result in abortion, stillbirth, uterine growth retardation, and low birth weight in the newborn. The purpose of this study was to assess the effects of HEI on knowledge and adherence to intermittent preventive treatment of malaria among pregnant women at secondary health facilities in Benue State, Nigeria. **Methods:** This quasi-experimental study included pre-, intervention, and post-intervention. The study recruited 871 pregnant women (436 study and 435 control) using multistage sampling. The study used a semi-structured questionnaire (pre- and post-test), follow-up checklist, and health education module. Participants self-administered the semi-structured questionnaire with 57 open-ended and closed-ended questions. **Results**: About 41% had high malaria awareness, but 93.9% did throughout pregnancy and intermittent preventive treatment (IPT) after health education intervention (HEI). The majority (93.8%) understood malaria transmission methods after HEI. 95.3% understood malaria symptoms after HEI. The HEI shows 95.6% of participants knew a lot about malaria during pregnancy. Post-HEI, 95% knew malaria prophylaxis. After HEI, 95.4% knew malaria-prevention drugs. Intermittent Preventive treatment (IPT) pregnancy dosages were known by 94.3% of respondents post-HEI. Post-HEI, 95.4% of responders knew the interval between IPT dosages, compared to 59.2% pre-HEI. After HEI, 95% of respondents were aware of IPT adverse effects, up from 29.2% pre-HEI. Pre-HEI showed. **Conclusions:** Results demonstrate HEI promotes malaria IPT adherence during pregnancy. A health education proves a veritable interventional strategy in influencing a mother’s understanding of malaria IPT, level of adherence to IPT, and drug adherence to directly observed therapy of IP while pregnant. Thus, nurses and midwives should increase IPT health education during antenatal clinic visits to increase its uptake and adherence among pregnant women and reduce malaria burden and death. Sulfadoxine/pyrimethamine (SP) for malaria in pregnancy (MiP) IPT must be distributed by the state health ministry to all health facilities—tertiary, secondary, primary, faith-based, and private.

## 1. Introduction

Malaria during pregnancy is a disease that exacerbates pregnancy complications such as anemia, spontaneous miscarriages, stillbirth, low birth weight, intrauterine growth restriction (IUGR), preterm birth, fetal distress, and congenital malaria [1]. Globally, WHO [2] reported an expected 241 million malaria cases and 627,000 fatalities in 2020. The WHO African Region bears a disproportionately large share of the global malaria burden, accounting for 95% of all cases and 96% of deaths [2]. Malaria accounts for more than 80% of infectious illness deaths in Africa, ranking second only to HIV/AIDS [3].

Infected pregnant women in malaria-endemic areas are generally asymptomatic, necessitating a preventive strategy [4]. The World Health Organization (WHO) recommends a three-pronged approach to preventing and treating MiP, which includes intermittent preventive treatment (IPT) with sulfadoxine/pyrimethamine, promotion and distribution of long-lasting insecticidal nets (LLINs) to pregnant women, and timely diagnosis and effective treatment of malaria cases and maternal anemia [5,6]. IPT with SP has been shown to reduce maternal anemia, low birth weights, and perinatal death in sub-Saharan Africa [7]. Malaria prevalence in Nigeria varies from 2% in Lagos to 52% in Kebbi State (NDHS, 2018). Although malaria prevalence in Nigeria has decreased from 42% in 2010 to 27% in 2015 and 23% in 2018, significant efforts are necessary to attain the 90% reduction by 2030 envisaged by the Sustainable Development Goal SDG 3 and eventually a malaria-free Nigeria [8].

The Nigerian Federal Ministry of Health has suggested that pregnant women receive intermittent preventative therapy (IPT) for malaria during pregnancy with sulfadoxine/pyrimethamine (SP). Pregnant women receive SP for free through prenatal care (ANC) services in public health and non-governmental facilities in compliance with national protocol. The National Demographic Health Survey [8] recommends that all women take three doses of sulfadoxine/pyrimethamine SP throughout pregnancy using the directly observed therapy (DOT) strategy.

Malaria infection during pregnancy is a major public health issue in sub-Saharan Africa. According to estimates, 97% of Nigeria’s population is susceptible to malaria infection, with children under the age of five and pregnant women being the most vulnerable to disease and mortality [7,9]. Nigeria has developed various programs and plans to reduce malaria. The worldwide Technical Strategy for Malaria 2016–2030 seeks to reduce worldwide malaria incidence and fatality rates by at least 90% by 2030, in line with Sustainable Development Goal 3 [8]. Furthermore, the announcement of the national malaria policy in February 2015 revealed a commitment to reaching a malaria-free Nigeria, addressing malaria prevention concerns [8]. The National Malaria Strategic Plan seeks to reduce malaria fatalities to less than 50 per 1000 births and parasite prevalence to less than 10% [9,10].

In 2001, the WHO proposed IPT with pyrimethamine/sulfadoxine as an MiP prevention method, which Nigeria’s government implemented in 2005 [11,12]. Pregnant women should receive three or more doses of IPT. However, IPT compliance in pregnancy has been low in Nigeria, notably in North Central Nigeria (PMI, 2019). According to the NDHS [8], 64% of pregnant women in Nigeria received a single dosage of IPT, 40% received two doses, and only 17% received three doses. IPT reduces placental parasitemia and protects against low birth weight in infants. IPT can reduce the number of clinical malaria episodes in infants and children. It also reduces the risk of anemia by preventing and treating new malaria infections and also reduces the risk of hospital admission in infants and children [13]. Sulfadoxine/pyrimethamine is effective against Plasmodium falciparum in pregnancy and infancy [14]. Although the effectiveness of IPTp-SP is challenged by the increasing resistance of parasite strains to SP. In response, the WHO is evaluating alternative drugs for IPTp [6].

IPT treatment common adverse effects include increased skin sensitivity to sunlight, tongue irritation or soreness, skin rash, stomach-ache or fever, chills, sore throat, swollen tongue, joint pain, cough, and shortness of breath in mothers [15]. Figueroa-Romero (2022) further observed that sulfadoxine and pyrimethamine are excreted in breast milk, which may cause jaundice and hemolytic anemia in the newborn, and therefore, sulfadoxine is contraindicated in pregnant women at term and breast-feeding mothers.

Malaria in pregnancy remains in Benue State, despite multiple initiatives by the state’s Ministry of Health to combat it. The condition continues to disproportionately affect pregnant women, resulting in unfavorable pregnancy outcomes [16]. According to data from the Benue State Hospitals Management Board (HMB, HMIS, 2019), of the 2028 pregnant women who attended ANC in 2019, 561 (27%) received one dosage of IPT, 462 (23% of the pregnant women) received two doses, and just 103 (5%) received three doses. Health education may be effective in increasing malaria knowledge and adhering to the IPT to minimize maternal morbidity and mortality, particularly in a malaria-endemic setting such as Nigeria. The aim of this study was to assess the impact of a health education intervention on knowledge and adherence to intermittent preventative malaria medication among pregnant women visiting ANC at secondary health facilities in Benue State, Nigeria. We hypothesize that providing health education interventions will have no significant impact on pregnant women attending ANC’s understanding of intermittent preventative malaria therapy.

## 2. Materials and Methods

### 2.1. Research Design

This study utilized a quasi-experimental approach, employing a pre-test–post-test design that included three stages: pre-intervention, intervention, and post-intervention. This design identified a control group (comparison group) that was similar to the intervention group (study group) in terms of baseline (pre-intervention) characteristics. The control group recorded the results without the implementation of a health education intervention. On the other hand, the study group received an intervention to test its effect on the treatment. We compared differences within and between groups to determine the impact of the health education intervention on IPT. We provided a health education intervention for malaria in pregnancy to the study group. The behavioral change communication (BCC) skill approach guided the health education intervention sessions. The control group did not receive any health education about IPT for malaria in pregnancy prior to the post-test. We conducted health education for the mothers in the control group following the post-tests to ensure ethical considerations in human research.

### 2.2. Study Area

The study was conducted in six selected secondary health care facilities across the three senatorial zones, which comprised zones A, B, and C in Benue State. Nigeria created Benue State, one of its 36 states, in 1976. Benue State has two distinct climatic seasons, the wet/rainy season and the dry/summer season. Temperatures fluctuate between 23 and 37 degrees Celsius across the year. The south-eastern part of the state adjoining the Obudu-Cameroun Mountain range, however, has a cooler climate similar to that of the Jos Plateau. The vegetation of the state consists of rain forests, which have tall trees, tall grasses, and oil palm trees that occupy the state’s western and southern fringes, while the Guinea savannah is found in the eastern and northern parts with mixed grasses and trees that are generally of average height. This vegetation appears to facilitate mosquito breeding. Benue State has about 1427 health facilities, which include public health facilities, faith-based health facilities, and private health facilities. Among the public health facilities are 23 secondary health care facilities (general hospitals) and two tertiary hospitals. Zones A and B have seven secondary health care facilities, respectively, while Zone C has nine secondary health care facilities. We used a simple random ballot technique to sample two general hospitals from each senatorial zone. Zone “A” includes general hospitals, Adikpo and Lessel; Zone “B” includes general hospitals, Makurdi and Wannune; while Zone “C” includes general hospitals, Otukpo and Okpoga. The geographic spread of the selected health facilities is approximately 100 km from each one.

### 2.3. Study Population and Sample

The target population consisted of 3032 pregnant women who were attending ANC at six selected secondary health facilities in Benue State. We applied a sample size formula, adopted from Fox, Hunn, and Mothers [17], to compare proportions in two equally sized groups.$$n=\frac{\left[P_{1}\left(1-P_{1}\right)+P_{2}(1-P_{2})\right]}{(P_{1}-P_{2}{)}^{2}}x\;K$$
where *P*_1_ = expected sample proportion for study group = 95%

*P*_2_ = expected sample proportion for control group = 90%

*K* = a constant which is a function of α and β 

α = 0.05

β = 0.2

Thus,$$\begin{array}{c} n=\frac{\left[0.95\left(1-0.95\right)+0.90(1-0.90)\right]}{(0.95-0.90{)}^{2}}x\;7.9\\ =\frac{\left[0.95\left(0.05\right)+0.90(0.1)\right]}{{\left(0.05\right)}^{2}}x\;7.9\\ =\frac{\left[\left(0.0475)+(0.09\right)\right]}{0.0025}x\;7.9\\ =\frac{0.1375}{0.0025}x\;7.9\\ =55x7.9\\ =434.5\\ =435 \end{array}$$

Therefore, *n* = 435 pregnant mothers.

The study used multistage sampling to recruit 871 pregnant women (436 study group and 435 control group). The first stage involved selecting all three senatorial districts in the state for the study. From each zone, two local governments were randomly selected by balloting. Stage II involved the purposeful selection of one GH from each of the selected LGs. The purposeful selection of hospitals was based on proximity or distance apart such that one hospital is at least 100 km apart from the other to avoid contamination of research information by respondents. The next stage involved the sampling of pregnant women in each of the six selected GHs. We computed the proportion of each sample in the respective health facilities by dividing the total representative population in each selected facility by the total number of all facilities, then multiplying this result by the number of patients in each facility to estimate the number of patients per health facility. All pregnant women who met the criteria for selection were all recruited as they presented themselves for antenatal care on a weekly basis from February 2020 to May 2020 until the desired number for each facility was obtained.

### 2.4. Data Collection Instruments

Data collection utilized a semi-structured questionnaire (pre- and post-test), a follow-up checklist, and a health education module. The participants self-administered the semi-structured questionnaire, which included 57 open-ended and closed-ended questions. The demographic characteristics of mothers include the mother’s age, educational qualification, marital status, tribe, and religion. Obstetric data include the number of pregnancies, the number of children alive, the number of children dead, the number of abortions, the duration of the current pregnancy before delivery, the occurrence of acute malaria attacks during pregnancy, the occurrence of anemia during pregnancy, the gravida status, and the IPT-SP dose received by the mothers. The IPT measures the mothers’ level of knowledge about malaria. IPT has 13 multiple-choice questions designed for assessment of the level of knowledge of mothers about malaria and intermittent preventive treatment of malaria in pregnancy. We assessed the participants’ knowledge level using a scoring system: good knowledge (2), average knowledge (1), and poor knowledge (0). We calculated the scores as follows: below 50% indicated poor knowledge, 51–59% indicated average knowledge, and above 60% indicated good knowledge. The mothers’ level of IPT adherence was coded and scored as follows: We assessed the mothers’ level of drug adherence during pregnancy as follows: good adherence (3–6 IPT doses), poor adherence (1–2 IPT doses), and no adherence (nil IPT dose).

The self-developed follow-up sheet (proforma) was used to obtain follow-up data on mothers IPT adherence during subsequent ANC visits. It consisted of 15 items. Similarly, the study groups designed a self-developed health education module to direct the delivery of health education interventions on malaria in pregnancy and IPT to pregnant women attending ANC. The health education module contained information about the program as follows: date of implementation and duration (30 min). Pregnant mothers make up the learner group, and the lecture room serves as the antenatal clinic. This module also lists the topics that need to be talked about. These include what malaria is, the organism that causes it, how it is transmitted, the risk factors for getting malaria, the signs and symptoms of malaria in pregnancy, complications of malaria, the best drug (sulfadoxine/pyrimethamine) for IPT of malaria, the dose schedule for IPT, ANC contacts by direct observed therapy (DOT), and how important it is to follow the IPT protocol. The module also outlined the learning objectives and instructional materials for use. Participants had the opportunity to ask questions and seek clarifications at the end of the HEI.

### 2.5. Validity of the Tools

In establishing the face and content validity of the tools, the research tools and research objectives were submitted to a jury of two experts in the Department of Nursing Sciences and one expert from the Faculty of Allied Health Sciences, College of Medical Sciences, Ahmadu Bello University Zaria. Given that the judgmental approach to content validity necessitates the presence of researchers alongside experts to ensure effective discussion, The judgmental approach to establishing face and content validity involved thorough literature reviews and follow-up by experts or panels. A content validity survey was then generated, and each item was assessed, and the survey was sent to the experts in the same field of the research. The instruments were critically assessed for relevance of content, clarity of statement, feasibility of the instrument, consistency of style, and logical accuracy. Their observations and corrections were used to modify the instruments and arrive at final copies of the instruments. To test for the reliability of the tools, a pilot study was conducted with mothers attending the antenatal clinic at the Family Support Program, Makurdi. This was done to determine the validity and appropriateness of the tools, as well as the index of stability (internal consistency). The coefficient of the internal consistency measure for the tools was 0.85, as determined by Cronbach’s alpha.

### 2.6. Data Collection Procedure

The data collection was performed in three phases. The first phase involved the selection and training of research assistants. The medical officer in charge of each hospital approached each of the six selected facilities to solicit permission to conduct the study in their respective facilities. Six trained research assistants were selected for each facility to assist in the data collection. Training of research assistants covered the study objectives, the use of a health education module for ANC women in the study group during the intervention phase, the administration of pre- and post-tests, and the use of a follow-up sheet. In pre-intervention (second phase), pregnant mothers who presented for booking at antenatal care were sampled after a clear explanation of the aim and nature of the study. Participants were informed about their right to participate or refuse or withdraw from participating in the study. Women who met the study inclusion criteria were recruited to participate in the study. At the study groups (General Hospital Lessel, General Hospital North Bank, Makurdi, and General Hospital, Otukpo), pre-test questionnaires were administered to the ANC women at booking. Pregnant mothers in the control groups, General Hospital, Adikpo, General Hospital, Wannune, and General Hospital Okpoga, also underwent pre-tests at the time of booking. In the intervention phase, health education intervention was administered using the health education module, while health education intervention was not administered in the control groups. In line with the behavioral change communication strategy, the trained research assistants also assisted with follow-up ANC visits to reinforce the health educational intervention, using the self-developed health educational training module. At the post-intervention phase, post-test questionnaires were administered. Post-tests were administered at a fixed ANC visit for each respondent during the third trimester of pregnancy.

Each respondent in the control groups received post-test questionnaires at a fixed ANC visit during the third trimester of pregnancy, followed by health education about IPT, to ensure ethical consideration for human subjects. Pregnant mothers in both the study group and the control group received a questionnaire to assess their perceived barriers to IPT adherence, as posited in the health belief model. We analyzed and compared the post-test results for both the study group and the control group, ensuring they aligned with the study objectives and test hypotheses.

### 2.7. Ethical Considerations

Ethical sanction was obtained from the Ethics and Research Committee of the Benue State Hospitals Management Board, Makurdi, ref. No. HMB/OFF/215/VOL/II/282, prior to the research’s commencement. A letter of ethical approval was submitted to the medical officer responsible for each facility prior to the commencement of the investigation. Informed consent was obtained from each participant. Respondents were guaranteed that their responses would be handled with the utmost privacy, anonymity, and confidentiality, and that their fundamental human rights would not be infringed upon. Furthermore, the respondents were granted the option to withdraw from the study at their discretion. Participants were informed of the purpose and nature of the study.

### 2.8. Data Analysis

We analyzed the data using descriptive (frequency counts, percentages, mean, and standard deviation) and inferential (chi-squared test) statistics. We accepted any positively worded item with a mean rating of 2.5 (aggregated mean score) and above as evidence of mothers’ high level of malaria knowledge and adherence to intermittent preventive treatment during pregnancy. However, we rejected items with a mean rating less than 2.5. We applied the reverse interpretation to the negatively worded items. We tested the research hypotheses at a 5% level of significance using a paired *t*-test. We used a paired *t*-test to ascertain the mothers’ level of knowledge about malaria during pregnancy and their adherence to intermittent preventive treatment (IPT). A χ^2^ test was used for categorical variables, respectively. All analyses were conducted using the Statistical Package for the Social Sciences (SPSS) 25.0, IBM SPSS, Chicago, IL, USA). A *p*-value less than 0.05 was considered statistically significant.

## 3. Results

Table 1 shows pre- and post-intervention malaria in pregnancy and IPT knowledge. Out of the 436 respondents, 179 (41%) demonstrated a high level of malaria knowledge, whereas 408 (93.9%) demonstrated this knowledge during pregnancy and during IPT after HEI. In the study group, 17 (4.0%) had a good understanding of malaria transmission techniques before HEI, whereas after HEI, 408 (93.8%) did. Pre-HEI data suggest that 44 (10.0%) of the study group knew malaria symptoms well. Post-HEI results for the same study group show 415 (95.3%) knew malaria symptoms well.

Only 24 individuals (5.6%) had a comprehensive understanding of malaria pregnancy complications prior to the HEI. The HEI shows that 416 people (95.6%) knew a lot about malaria problems in pregnancy. The distribution on malaria IPT knowledge was as follows: 413 (95%) had good malaria prevention knowledge post-HEI, while 43 (9.8%) did during pre-HEI. In the pre-HEI distribution of knowledge regarding malaria-prevention medicine, 227 (52.0%) respondents had good knowledge. The same study group found 415 (95.4%) with good knowledge after HEI. Before the HEI, 232 (53.3%) people understood pregnant IPT dosages well. The same study group’s post-HEI results showed that 410 (94.3%) respondents knew IPT doses for pregnancy, while the rest did not. One hundred and eighty-three (42.0%) respondents had good pre-HEI understanding of IPT’s benefits to mothers. Post-HEI showed 416 (95.6%) respondents had good knowledge. Pre-test replies for IPT advantages for the baby suggest 26 (6.0%) had good knowledge. Post-HEI showed 410 (94.3%) responders had good knowledge. Pre-HEI showed that 258 (59.2%) respondents knew the interval between the first and subsequent IPT dosages, but post-HEI demonstrated that 415 (95.4%) did. Pre-HEI, only 127 (29.2%) respondents had strong awareness of IPT side effects, while post-HEI, 413 (95%) did. Pre-HEI showed that 193 (44.3%) responders knew how to handle IPT side effects. Post-HEI, 416 (95.7%) respondents had good knowledge. With pre-HEI awareness of additional malaria prevention methods, 40 (9.2%) were knowledgeable. Post-HEI results showed 415 (95.4%) respondents had strong awareness of malaria prevention methods.

Before and after the health education intervention, expectant mothers’ knowledge significantly differed between low, average, and good levels (Table 2).

Table 3 shows study group awareness of malaria in pregnancy and IPT before and after HEI. The study group’s mean level of good knowledge was 28% before HEI. However, following HEI, the same study group had a 95% mean knowledge score. There was a statistically significant difference between pregnant mothers’ knowledge before and after HEI in the study group. The study group’s expectant mothers, both with poor and average knowledge, did not show any significant differences before and after HEI.

Table 4 shows both groups’ post-intervention IPT adherence during pregnancy (at least 3 doses). In the study, 415 (95.1%) respondents followed IPT completely. The control group showed 107 (24.6%) total IPT adherence after HEI. The majority of respondents, 413 (94.8%), reported complete adherence to the IPT during their last pregnancy (excluding primigravida), while only 104 (23.8%) in the control group did so without HEI. IPT under DOT showed that 414 (95%) respondents followed it thoroughly. In contrast, 108 (24.8%) control post-intervention participants followed IPT administration under DOT.

Table 5 shows post-intervention adherence rates for mothers in the two groups. The non-HEI control group had a mean complete adherence rate of 33%. Following HEI, the study group had a mean full adherence rate of 95%. The chi-square analysis showed no significant difference in pregnant women’s adherence (1–2 doses) after health education. However, expectant mothers’ adherence for three doses and above differed significantly between the study and control groups.

Table 6 shows the mean (1.10) for pre-test scores of the mother’s levels of knowledge and drug adherence to IPT before HEI and the mean (1.59) for post-test scores of the mother’s level of knowledge and IPT adherence throughout pregnancy, respectively. The results also revealed a t-value of −10.865 and a *p*-value of 0.000, indicating the rejection of the null hypothesis (H0), which suggests that mothers’ knowledge levels affect their adherence to IPT during pregnancy.

## 4. Discussion

The current study found that only a small minority (28.0%) of ANC pregnant women had a thorough awareness of malaria prior to HEI, while the majority (95%) of the women did after HEI. This shows that HEI greatly improves their understanding about malaria during pregnancy. Adamu [18] discovered a similarly low level of malaria knowledge (52.6%) among pregnant women visiting ANC in Jigawa State, Nigeria; nevertheless, IPT training for health personnel was suggested. Yaya et al. [19] found that women in Burkina Faso had an average knowledge (56.1%) and advocated behavioral change, such as communication interventions, to improve overall malaria awareness. The current study’s findings, however, contrast those of Singh et al. [20], who discovered that 90% of rural people in Aliero, Northern Nigeria, had comprehensive knowledge about malaria control strategies. However, their study observed that knowledge about preventive measures does not necessarily translate into improved practice; hence, they recommended a health education intervention. The study’s findings showed a significant improvement in all pregnancy-related malaria indicators of interest. The findings are consistent with those of Ayiisi [21], who evaluated ANC women’s knowledge and usage of IPT for malaria control in Sunyani West District, Ghana. The study found that pregnant women had relatively little awareness of IPT. The majority of respondents (68.2%) were unfamiliar with IPT use. The study advised that women receive more health education about the use of IPT through various media, including radio and television.

The current study found that 95% of ANC pregnant women adhered completely to IPT after HEI, compared to 53% before HEI. This demonstrates the effectiveness of HEI. Adamu [18] discovered substantially poorer findings on adherence to IPT among pregnant women visiting ANC in Jigawa State, Nigeria, revealing that the majority of the pregnant women (80.0%) had inadequate awareness of IPT and very low adherence (1.4%) to two or more doses of IPT. This points to the fact that there is a need for intensified health education intervention to improve IPT adherence among the pregnant women. Ayiisi [21] conducted a study to assess the knowledge and utilization of IPT for malaria control among pregnant ANC women in Sunyani West District, Ghana, and found significantly low adherence or utilization of IPT among these women. The remarkable improvement of the knowledge of the IPT drug to prevent malaria in pregnancy, its doses during pregnancy, benefits to both the mother and the baby, the interval for the first dose and its subsequent doses, side effects, and the steps to follow when experiencing side effects from IPT indicate HEI influence on the knowledge of pregnant women about malaria during pregnancy. Therefore, healthcare providers should intensify their educational intervention efforts on IPT for malaria in pregnancy through various media to prevent malaria infection.

The study found that following HEI, women undergoing ANC had a high understanding of malaria IPT (95.0%). This finding clearly indicates that HEI has a good influence on women. The current study’s findings are consistent with Ahmed’s [22] study on the influence of a health education initiative (HEI) on malaria knowledge in Northeast Nigeria. This study found that the HEI considerably increased pregnant women’s knowledge, resulting in a 12.75% rise in total knowledge scores for the intervention group against the control group. The Oyefabi et al. [23] study in Nigeria reported similar findings on the effect of primary health care workers training on the knowledge and utilization of intermittent preventive therapy for malaria. During the baseline assessment of the study group, only 11.8% of the respondents had good knowledge of the IPT. This, however, increased significantly to 87.4% of clients’ post-intervention. The study showed a significant improvement in the IPT knowledge of the study group’s clients compared to the control group, which did not receive such training. According to the Owusu-Addo et al. [24] study in Kumasi, Ghana, health education interventions (HEIs) are effective and remain a valuable tool in community-based malaria prevention and control interventions. HEIs influence the uptake of community-based malaria prevention and control interventions, enhance knowledge about malaria, and generally reduce malaria prevalence and mortality among pregnant women.

According to the current study, IPT adherence by women visiting ANC increased (95.0%) after HEI, compared to 33% of ANC women having total adherence before HEI. Furthermore, HEI had a substantial effect on the mother’s adherence to IPT while pregnant. This implies that the HEI had a beneficial impact on the mother’s adherence to a minimal number of IPT doses, which is consistent with the WHO’s [25] recommendation that all pregnant women receive IPT and at least three doses of SP during their pregnancy. Badirou [26] emphasized that raising awareness and enhancing women’s knowledge of malaria is critical to achieving IPT adherence. Communication is essential for promoting social and behavioral changes among pregnant women, family decision-makers, community leaders, and healthcare providers.

The majority of mothers have completed at least three doses of IPT. As a result, 5% of patients began IPT but did not complete the required three doses. This small number of pregnant women appears to have important implications for the WHO’s 2012 target, which states that a pregnant woman should receive at least three SP doses during her pregnancy. Furthermore, this contradicts the Nigerian national malaria policy, which was launched in February 2015 and underlined the government’s commitment to eradicating malaria at all levels. Adewuyi et al. [27] discovered that pregnant women who did not complete IPT were less likely to seek ANC services. In Nigeria, 46.5% of pregnant women did not receive ANC services, whereas 61.1% resided in rural regions and 22.4% lived in cities. The north-west area of Nigeria exhibited the highest prevalence of ANC underutilization, with percentages of 69.3%, 76.6%, and 44.8% for both rural and urban homes. In Uganda, Mbonye [28] discovered that in order to boost adherence to IPT, it is critical to raise awareness and train midwives about the benefits and importance of SP, as well as adherence to two doses of IPT for all pregnant women during prenatal care. Conn [29] defined non-adherence as situations in which a patient does not commence therapy, fails to implement but delays, omits, or takes extra doses, or initiates but discontinues treatment.

## 5. Conclusions

The study’s findings showed that HEI remains an effective technique for improving adherence to IPT for malaria during pregnancy. A health education intervention had a substantial influence on the mother’s understanding of malaria IPT, level of adherence to IPT, and drug adherence to directly observed therapy of IP while pregnant. Future studies should endeavor to explore other intervention measures in assessing the effectiveness of intermittent preventive malaria treatments among pregnant women. Furthermore, there is a substantial association between mothers’ level of understanding and their adherence to IPT during pregnancy. As a result, nurses and midwives should increase their health education intervention efforts on IPT during antenatal clinic visits to promote its uptake and adherence among pregnant women, lowering the malaria burden and malaria-related mortality in the context. The State Ministry of Health must ensure the timely and consistent supply of sulfadoxine/pyrimethamine (SP) for IPT of MiP to all health facilities in the state, including tertiary, secondary, and primary health care, faith-based, and private health facilities.

## Figures and Tables

**Table 1 healthcare-13-00105-t001:** Pre- and post-intervention levels of knowledge.

Variables	Study Group Before (n = 436)					Study Group After (n = 436)			Statistical Test of sig.	
	Poor Knowledge	Average Knowledge	Good Knowledge	χ^2^	*p*-Value	Poor Knowledge	Average Knowledge	**Good Knowledge**	**χ^2^**	** *p* ** **-Value**
	n (%)	n (%)	n (%)			n (%)	n (%)	**n (%)**		
Meaning of malaria	44 (10.0)	214 (49.0)	179 (41.0)	1360.832	<0.000	4 (0.9)	23 (5.2)	408 (93.9)	2291.881	<0.001
Mode of transmission	148 (34.0)	270 (62.0)	17 (4.0)	1167.880	<0.000	3 (0.6)	24 (5.6)	408 (93.8)	1880.197	<0.001
Signs and symptoms of MiP	140 (32.0)	253 (58.0)	44 (10.0)	2341.600	<0.000	3 (0.6)	18 (4.1)	415 (95.3)	2688.156	<0.001
Complications of MiP	139 (31.8)	273 (62.6)	24 (5.6)	787.963	<0.000	1 (0.3)	17 (4.0)	416 (95.6)	2472.506	<0.001
Methods of preventing MiP	140 (32.1)	253 (58.0)	43 (9.8)	1445.931	<0.000	4 (1.0)	17 (4.0)	413 (95.0)	1985.759	<0.001
Drugs used to prevent MiP	39 (9.0)	170 (939.1)	227 (52.0)	2304.408	<0.000	3 (0.8)	17 (3.8)	415 (95.4)	1856.680	<0.001
Doses of malaria drug during MiP	110 (25.3)	93 (21.4)	232 (53.3)	1198.083	<0.000	7 (1.5)	18 (4.2)	410 (94.3)	2527.806	<0.001
Benefits of SP-IPT to the mother	140 (32.0)	113 (26.0)	183 (42.0)	1277.720	<0.000	3 (0.6)	17 (3.8)	416 (95.6)	2877.822	<0.001
Benefits of SP-IPT to the baby	135 (31.0)	275 (63.0)	26 (6.0)	837.275	<0.000	3 (0.6)	22 (5.1)	410 (94.3)	2094.404	<0.001
When should pregnant mothers take 1st dose of IPT and subsequent doses	78 (18.0)	99 (22.8)	258 (59.2)	2072.163	<0.000	2 (0.5)	18 (4.1)	415 (95.4)	1893.529	<0.001
Side effects of IPT	175 (40.2)	133 (30.6)	127 (29.2)	1678.148	<0.000	4 (0.9)	18 (4.1)	413 (95.0)	2582.459	<0.001
Steps to take when side effects from IPT are experienced	59 (13.6)	184 (42.2)	193 (44.3)	2295.671	<0.000	4 (0.9)	15 (3.4)	416 (95.7)	1681.590	<0.001
Other measures used to prevent mosquito bites	145 (33.2)	252 (57.7)	40 (9.2)	1158.410	<0.000	3 (0.6)	17 (4.0)	415 (95.4)	2926.407	<0.001

**Table 2 healthcare-13-00105-t002:** Pre- and post-intervention mean knowledge scores of study group.

Variables	Pre-Test	Post-Test	t *p*
Meaning of malaria	1.31	1.93	−11.472, 0.000
Mode of transmission	0.70	1.93	
Signs and symptoms of malaria in pregnancy	0.78	1.95	
Complications of malaria in pregnancy	0.74	1.95	
Methods of preventing malaria in pregnancy	0.78	1.94	
Drugs used to treat and prevent malaria in pregnancy	1.43	1.95	
Doses of malaria drug during pregnancy	1.28	1.93	
Benefits of IPT for the mother	1.10	1.95	
Benefits of IPT for the baby	0.75	1.94	
When should pregnant mothers take first dose of IPT and subsequent doses of IPT	1.41	1.95	
Side effects of IPT	0.89	1.94	
Steps to take when side effects from IPT are experienced	1.31	1.95	
Other measures used to prevent mosquito bites	0.76	1.95	
Statistical test of significance	χ^2^p =		19.810, 0.047

**Table 3 healthcare-13-00105-t003:** Mother’s knowledge about malaria in pregnancy IPT before and after in the study group.

Knowledge Status	Study Before (n = 436)	Study After (n = 435)	χ^2^	*p*-Value
	n (%)	n (%)		
Poor <50%	113 (26)	4 (1.0)	192.000	0.291
Average 50–60%	201(46)	22 (5.0)	208.000	0.265
Good >60%	122 (28)	413 (95.0)	112.000	0.003

**Table 4 healthcare-13-00105-t004:** Participants level of adherence to IPT after HEI.

Variables	Post-Test Study (n = 436)			Post-Test Control (n = 435)			χ^2^	*p*-Value
	No Adherence (No Doses)	Poor Adherence (1–2 Doses)	Complete Adherence (3–6 Doses)	No Adherence (No Doses)	Poor Adherence (1–2 Doses)	Complete Adherence (3–6 Doses)		
	n (%)	n (%)	n (%)	n (%)	n (%)	n (%)		
Doses of IPT taken during present pregnancy	7 (1.6)	14 (3.3)	415 (95.1)	27 (6.3)	301 (69.1)	107 (24.6)	42.000	0.240
Doses of IPT taken during last pregnancy	6 (1.4)	17 (3.8)	413 (94.8)	38 (8.7)	294 (67.5)	104 (23.8)	35.000	0.110
IPT administered under DOT	4 (0.9)	18 (4.1)	414 (95.0)	33 (7.7)	294 (67.5)	108 (24.8)	35.000	0.110

**Table 5 healthcare-13-00105-t005:** Mother’s level of adherence after intervention.

Adherence Status	Study (n = 436)	Control (n = 435)	χ^2^	*p*-Value
	n (%)	n (%)		
No adherence (no dose of IPT taken)	4 (1.0)	26 (6.0)	9.330	0.079
Poor adherence (1–2) dose	17 (4.0)	265 (61.0)	11.660	0.053
Complete adherence > 3 dose	414 (95.0)	144 (33.0)	12.44	0.046

**Table 6 healthcare-13-00105-t006:** Relationship between mother’s level of knowledge and adherence to IPT during pregnancy.

Paired Samples Statistics											
		Mean		N		Std. Deviation			Std. Error Mean		
Pair 1	Pre-test scores knowledge and adherence	1.10		96		0.331			0.034		
	Post-test scores knowledge and adherence	1.59		96		0.379			0.039		
Paired samples test											
		Paired differences							t	df	Sig. (2-tailed)
		Mean	Std. Deviation		Std. Error Mean		95% Confidence Interval of the Difference				
							Lower	Upper			
Pair 1	Pre-test scores knowledge and adherence—post-test scores knowledge and adherence	−0.493	0.445		0.045		−0.583	−0.403	−10.865	95	0.000

## Data Availability

Data available on request.

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
