# Peer review of "Effect of Health Education Intervention on Knowledge and Adherence to Intermittent Preventive Treatment of Malaria in Pregnancy Among Women"

_healthcare, 2025, doi:10.3390/healthcare13020105_

Round 1
Reviewer 1 Report
Comments and Suggestions for Authors
Dear Authors,
I found the overall paper interesting and good but the Discussion part needs a thorough review and revision in addition to some other minor changes in the methods and results. The detailed comments are attached.
Best wishes .

Dear editors,
The paper needs revision before publication. Changes are required for clarity in some parts of the text under different sections. Additionally, the discussion needs a thorough review and revision for meaningful comparisons and interpretation of the cited references. it will be good if the references are updated to add DOI where ever possible to keep up with the latest trends of publication.
Author Response
Overall Comments:
The objective of this study was to assess the effects of Health Education Intervention (HEI) on knowledge and adherence to intermittent preventive treatment of malaria among pregnant women at secondary health facilities in Benue State of Nigeria.
A quasi-experimental study was used wherein pre-intervention, and post-intervention data was compared for 871 pregnant women (436 study and 435 control) for which a semi-structured questionnaire (pre- and post-test), follow-up checklist, and health education module were used. Participants self-administered the semi-structured questionnaire with 57 open-ended and closed-ended questions. The analysis was carried out hypothesizing that the provision of health educational intervention will not significantly impact the knowledge of intermittent preventive treatment of malaria among pregnant women attending ANC.
Based on the results and analysis of the responses the authors conclude that i) HEI promotes malaria IPT adherence during pregnancy ii) a substantial link exists between mothers' understanding and IPT adherence during pregnancy iii) nurses and midwives should increase IPT health education during antenatal clinic visits to increase its uptake and adherence among pregnant women and reduce malaria burden and death iv) Sulphadoxine Pyramithamine (SP) for MIP IPT must be distributed by the state health ministry to all health facilities i.e., tertiary, secondary, primary, faith based, and private.
The overall paper is well written, but some changes are required in different sections of the paper, especially the Discussion, which needs to be reviewed, revised and worded appropriately.
Some specific comments are further provided below to improve the paper:
Abstract:
Check Line no 17 (Methods) & line no 21(Results) and revise to ensure that the intended meaning is conveyed appropriately.
Response: This has been affected to read:
This quasi-experimental study involved three phases: pre-intervention, intervention, and postintervention. Using multistage sampling, the study enrolled 871 pregnant women (436 study participants and 435 controls). The study utilized a semi-structured questionnaire (pre- and post-test), a follow-up checklist, and a health education module
Introduction:
Materials and Methods:
- Use full form of abbreviations like GH, LGs etc at least once in this section as well as rest of the paper initially with abbreviations in parenthesis ( ).
- Data collection utilized a semi-structured questionnaire (pre- and post-test), a follow-up check list, and a health education module.
- The semi-structured questionnaire (pre- and post-test), as well as the follow-up check list should be provided as supplementary data for reference.
Response:
The above issues have been addressed accordingly. The use of abbreviations has been defined first mentioning.
Results:
- In line no. 282 something is amiss. Check ‘while 43 (9.8%) did’ to ensure that the intended meaning is conveyed appropriately.
Response: This has been corrected:
The distribution on malaria IPT knowledge was as follows: 413 (95%) had good malaria prevention knowledge post-HEI, while 43 (9.8%) did during Pre HEI.
- It would be relevant to signify the change in knowledge post HEI for all results (in the text paragraphs explaining the results) in terms of percentage to make it easily.
- comprehendible by the readers s well as appreciate the % change for each studied parameter.
Discussion:
- The first two sentences (Lines 333-335) are contradictory. In subsequent sentences, References 15 and 16 seem to be misinterpreted the level of knowledge in both being above 50% which is much higher than 28%.
Response: The contradiction has been corrected to read:
Adamu [15] discovered a higher level of malaria knowledge (52.6%) among pregnant women visiting ANC in Jigawa State, Nigeria; nevertheless, IPT training for health personnel was suggested. Yaya et al. [16] found that women in Burkina Faso had an average knowledge (56.1%) and advocated behavioural change, such as communication interventions, to improve overall malaria awareness.
- Lines 342-343 need to be reviewed as the meaning is not clear , both appear to be contradictory statements.
The sentences have been corrected.
- Further the statements pertaining to Reference 18 are not quantified and comparable in terms of Percentages due to absence of this information in the discussion. Similarly the facts and figures compared in the next study look confusing - 28 % (present study) vs. 1.4 % (Ref 16). It is recommended that the statements written in the rest of the discussion comparing the results of the present study with the published ones are also reviewed and reworded appropriately, where ever necessary.
The paragraphed has been re-written as:
The current study's findings, however, contrast those of Singh et al. [16], who discovered that 90% of rural people in Aliero, Northern Nigeria had comprehensive knowledge about malaria control strategies. However, their study observed that knowledge about preventive measures does not necessarily translate into improved practice; hence, they recommended a health education intervention. The study's findings showed a significant improvement in all pregnancy-related malaria indicators of interest.
- Overall, the entire discussion needs to be reviewed and rewritten thoughtfully, especially the first three paragraphs, ensuring the relevance of facts to figures is maintained and interpretation is statistically correct.
Response: The entire discussion sections have been reviewed and corrections made.
References:
- Please ensure that all references are written maintaining a common style as per the journal format.
Response: All the references have been updated as corrected.
- Add DOI where ever available and relevant to keep up with the latest trends of publication.
Response: The journal does not in include DOI in the reference list as it is not part of their reference style.
Reviewer 2 Report
Comments and Suggestions for Authors
Comments 3343689
1. In the research paper titled “Effect of health education intervention on knowledge and adherence to intermittent preventive treatment of malaria in pregnancy among women” the author aimed to educate the women about malaria in pregnancy, its risks, and the benefits of preventive treatment. Also specifically demonstrated how the intermittent preventive treatment (IPT), which involves taking malaria medication at scheduled times during pregnancy as the malaria symptoms are asymptomatic during pregnancy. This research is important due to the significant global burden of treating malaria during pregnancy.
2. The manuscript is well-crafted and emphasizes the necessity of awareness and knowledge about malaria and its treatment in pregnant women. This education would help to avoid pregnancy complication, if pregnant women are infected with malaria.
3. However, the author should include the 2023 malaria report
4. The author should mention the side effects of IPT treatment.
5. The author should discuss the effect of IPT in babies after birth and whether the treatment prevents babies from future malaria infection.
6. The author should also discuss whether mothers and babies are facing any side effects after birth.
7. The author should mention the Sulphadoxine Pyramithamine (SP) specifically against which species of the malaria parasite.
8. The author should also mention SP could also be used for the treatment of drug-resistant parasites during pregnancy.
9. The author should enhance the presentation of results by using more visual methods, such as bar diagrams or graphs. This must be included.
10. The author should discuss the future direction of the study.
I recommend this article for acceptance after the author addresses all of the above points.
Author Response
REVIEWER 2
Comments and Suggestions for Authors
Comments 3343689
- In the research paper titled “Effect of health education intervention on knowledge and adherence to intermittent preventive treatment of malaria in pregnancy among women” the author aimed to educate the women about malaria in pregnancy, its risks, and the benefits of preventive treatment. Also specifically demonstrated how the intermittent preventive treatment (IPT), which involves taking malaria medication at scheduled times during pregnancy as the malaria symptoms are asymptomatic during pregnancy. This research is important due to the significant global burden of treating malaria during pregnancy.
The manuscript is well-crafted and emphasizes the necessity of awareness and knowledge about malaria and its treatment in pregnant women. This education would help to avoid pregnancy complication, if pregnant women are infected with malaria. However, the author should include the 2023 malaria report
The 2023 malaria report included as suggested: World health Organization (2023). Malaria Prevention and treatment. https://www.google.com/search?q=2023+m+report+on+Malaria+intermittent+preventive+treatment&oq=2023
- The author should mention the side effects of IPT treatment.
Response: This has been included to read: IPT treatment common adverse effects include increased skin sensitivity to sunlight, tongue irritation or soreness, skin rash, stomach-ache or fever, chills, sore throat, swollen tongue, joint pain, cough, and shortness of breath in moms. (Figueroa-Romero, 2022). Figueroa-Romero (2022) further observed that Sulfadoxine and pyrimethamine is excreted in breast milk which may cause jaundice and haemolytic anaemia in the newborn and therefore, Sulfadoxine is contraindicated in pregnant women.
- The author should discuss the effect of IPT in babies after birth and whether the treatment prevents babies from future malaria infection.
Response: The effect of IPT in babies has been discussed and added to read:
IPT reduces placental parasitemia and protect against low birth weight in infants. IPT can reduce the number of clinical malaria episodes in infants and children. It also re-duces risk of anemia by preventing and treating new malaria infections and also reduces risk of hospital admission in infants and children (Esu, 2019).
- The author should also discuss whether mothers and babies are facing any side effects after birth.
Response: Figueroa-Romero (2022) observed that Sulfadoxine and pyrimethamine is excreted in breast milk. Sulfonamides may cause jaundice and haemolytic anaemia in the newborn and is contraindicated in pregnant women at term or breast feeding mothers.
- The author should mention the Sulphadoxine Pyramithamine (SP) specifically against which species of the malaria parasite.
Response: This has been mentioned and included to read:
Sulfadoxine-pyrimethamine is effective against Plasmodium falciparum in pregnancy and infancy (Chaponda (2021).
- The author should also mention SP could also be used for the treatment of drug-resistant parasites during pregnancy.
This has been mentioned to read:
Although, the effectiveness of IPTp-SP is challenged by the increasing resistance of parasite strains to SP. In response, the WHO is evaluating alternative drugs for IPTp (WHO, 2023)
- The author should enhance the presentation of results by using more visual methods, such as bar diagrams or graphs. This must be included.
Thank so much for the suggestion
- The author should discuss the future direction of the study.
A health education intervention had a substantial influence on the mother's understanding of malaria IPT, level of adherence to IPT, and drug adherence to Directly Observed Therapy of IP while pregnant. Future studies should endeavour to explore other intervention measures in assessing the effectiveness of intermittent preventive malaria treatments among pregnant women.
I recommend this article for acceptance after the author addresses all of the above points.

Reviewer 3 Report
Comments and Suggestions for Authors
The authors conducted an important and essential study on a project of knowledge generation in pregnant women about intermittent preventative treatment.
Major comments: In the abstract and manuscript the authors should only mention in the conclusions what is backed up by the result sections of the abstract and main manuscript.
Detailed comments:
The authors should use the past tense throughout the manuscript. In the abstract the authors need to spell out the acronyms HEI and IPT at the first mentioning and put the acronyms in brackets behind the full terms.
The sentence: "A statistically significant difference occurs between pregnant mothers' adequate knowledge before and after health education interventions in the study group." needs to be rephrased to state: "There was a statistically significant difference between pregnant mother's knowledge before and after HEI in the study group" and this sentence needs to be moved to the first mentioning of percentages of knowledge.
Author Response
Comments and Suggestions for Authors
The authors conducted an important and essential study on a project of knowledge generation in pregnant women about intermittent preventative treatment.
Major comments: In the abstract and manuscript the authors should only mention in the conclusions what is backed up by the result sections of the abstract and main manuscript.
Response: The conclusion has been revised to align with the conclusion in the abstract and the conclusion in the main manuscript.
“The study's findings showed that HEI remains an effective technique for improving adherence to IPT for malaria during pregnancy. A health education intervention had a substantial influence on the mother's understanding of malaria IPT, level of adherence to IPT, and drug adherence to Directly Observed Therapy of IP while pregnant. Future studies should endeavour to explore other intervention measures in assessing the effectiveness of intermittent preventive malaria treatments among pregnant women. Furthermore, there is a substantial association between mothers' level of understanding and their adherence to IPT during pregnancy. As a result, nurses and midwives should increase their health education intervention efforts on IPT during antenatal clinic visits to promote its uptake and adherence among pregnant women, lowering the malaria burden and malaria-related mortality in the context. The State Ministry of Health must ensure the timely and consistent supply of Sulphadoxine Pyramithamine (SP) for IPT of MIP to all health facilities in the state, including tertiary, secondary, and primary health care, faith-based, and private health facilities”.
Detailed comments:
The authors should use the past tense throughout the manuscript. In the abstract the authors need to spell out the acronyms HEI and IPT at the first mentioning and put the acronyms in brackets behind the full terms.
Response:
The entire manuscript has been revised to address the comment regarding the use of past tense. Furthermore, the acronyms HEI and IPT are defined at first mentioning as suggested, See below and in the text: “About 41% had high malaria awareness, but 93.9% did throughout pregnancy and Intermittent Preventive Treatment (IPT) after Health Education Intervention (HEI)”.
The sentence: "A statistically significant difference occurs between pregnant mothers' adequate knowledge before and after health education interventions in the study group." needs to be rephrased to state: "There was a statistically significant difference between pregnant mother's knowledge before and after HEI in the study group" and this sentence needs to be moved to the first mentioning of percentages of knowledge.
Response:
The suggestion is followed and addressed thus:
Table 3 shows study group awareness of malaria in pregnancy and IPT before and after HEI. The study group's mean level of good knowledge was 28% before HEI. However, following HEI, the same study group had a 95% mean knowledge score. There was a statistically significant difference between pregnant mother's knowledge before and after HEI in the study group. The study group's expectant mothers, both with poor and average knowledge, did not show any significant differences before and after HEI.
